# Challenges in Toxicological Risk Assessment of Environmental Cadmium Exposure

**DOI:** 10.3390/toxics13050404

**Published:** 2025-05-16

**Authors:** Soisungwan Satarug

**Affiliations:** Centre for Kidney Disease Research, Translational Research Institute, Woolloongabba, Brisbane, QLD 4102, Australia; sj.satarug@yahoo.com.au

**Keywords:** cadmium, benchmark dose, dose–response, eGFR, NOAEL, threshold-based risk assessment

## Abstract

Dietary exposure to a high dose of cadmium (Cd) ≥ 100 µg/day for at least 50 years or a lifetime intake of Cd ≥ 1 g can cause severe damage to the kidneys and bones. Alarmingly, however, exposure to a dose of Cd between 10 and 15 µg/day and excretion of Cd at a rate below 0.5 µg/g creatinine have been associated with an increased risk of diseases with a high prevalence worldwide, such as chronic kidney disease (CKD), fragile bones, diabetes, and cancer. These findings have cast considerable doubt on a “tolerable” Cd exposure level of 58 µg/day for a 70 kg person, while questioning the threshold level for the Cd excretion rate of 5.24 µg/g creatinine. The present review addresses many unmet challenges in a threshold-based risk assessment for Cd. Special emphasis is given to the benchmark dose (BMD) methodology to estimate the Cd exposure limit that aligns with a no-observed-adverse-effect level (NOAEL). Cd exposure limits estimated from conventional dosing experiments and human data are highlighted. The results of the BMDL modeling of the relationship between Cd excretion and various indicators of its effects on kidneys are summarized. It is recommended that exposure guidelines for Cd should employ the most recent scientific research data, dose–response curves constructed from an unbiased exposure indicator, and clinically relevant adverse effects such as proteinuria, albuminuria, and a decrease in the estimated glomerular filtration rate (eGFR). These are signs of developing CKD and its progression to the end stage, when dialysis or a kidney transplant is required for survival.

## 1. Introduction

The conventional toxicological risk assessment for any health hazardous substance is reliant on dose–response curves constructed from a series of experimentation, which typically involves daily administration of four to five different doses for 90 days or longer [1,2,3,4,5]. The dose–response curves are used to define the lower bound “no-observed-adverse-effect level” (NOAEL) and the upper bound “lowest-observed-adverse-effect level” (LOAEL) from which a point of departure (POD) from normalcy is established. The POD value forms a basis to estimate health guidance values, which are known as the minimal risk level (MRL), the toxicological reference value (TRV), the reference dose (RfD), the tolerable weekly intake (TWI), and the tolerable monthly intake (TMI) [1,2,3,4,5,6]. These different terms create unnecessary confusion and stumbling blocks. They should be standardized so as to translate them to public health protection actions.

The NOAEL value derived from animal dosing experiments is the highest dose tested that produces a statistically insignificant effect compared to controls. The NOAEL value can be translated to the benchmark dose (BMD) with an inclusion of an uncertainty or safety margin to compensate for species differences and human variability [1]. To circumvent the animal-to-human extrapolation, the BMD approach has been applied to human population data [1,2,3,4,5]. In this way, the benchmark dose limit (BMDL) value has replaced the experimental NOAEL.

The POD-based health guidance values MRL, TRV, RfD, TWI, and TMI all rely on the premise that a critical exposure level exists above which an adverse effect cannot be discernable [1]. In effect, an exposure level derived from the most sensitive endpoint would be protective against all other adverse effects [1]. A notable limitation is that threshold-based risk assessment is only applicable to non-cancer endpoints [1,5]. An evaluation of the carcinogenic risk of a suspected entity requires an observation over a lifespan, such as a two-year rodent/murine bioassay [7,8,9].

The present review has its focus on the metal contaminant cadmium (Cd), which is found in most foodstuffs [10,11,12,13], tobacco smoke, and airborne particle pollution [14,15]. Because of the extremely slow excretion rate of Cd, the metal is retained within the cells of nearly all tissues and organs in the body, notably the kidney [16,17]. Concerningly, Cd is a human cancer-causing agent; epidemiological studies have linked an increased risk of tumors in the lungs, kidneys, pancreas, breast, and liver to chronic environmental Cd exposure [7,8,9,18]. These data are in line with a two-year bioassay that indicated Cd as a multi-tissue carcinogen [7]. The ability of Cd to induce non-tumorigenic human cells to undergo malignant transformation strengthens its oncogenic potential [18,19].

The first objective of this review is to discuss current health guidance values for Cd and highlight their inadequacy to provide protection against health risks associated with excessive exposure to Cd. It reiterates the total imprecision in measuring Cd exposure and/or associated adverse outcomes, which biases dose–response relationships toward the null [20]. The second objective is to illustrate non-differential errors imposed on datasets by adjusting Cd excretion (E_Cd_) to creatinine excretion (E_cr_) as E_Cd_/E_cr_. These errors can be eliminated by normalizing E_Cd_ to creatinine clearance (C_cr_) as E_Cd_/C_cr_. As a third objective, fundamental and practical knowledge on the BMD methodology is provided together with a summary of BMD modeling results to accentuate a continuing effort to identify a toxic endpoint that can form a reliable basis to formulate a meaningful health guideline value for Cd exposure.

## 2. Existing Dietary Cd Exposure Guidelines

This section provides health guidance values for exposure to Cd in the diet together with exposure threshold levels that have been estimated. Because nearly all food types contain Cd as a contaminant, exposure to Cd occurs through a normal diet [10,11,12,13], and foods which are consumed frequently in a large quantity, like staples, contribute the most to dietary Cd exposure [11].

The consumption of Cd at ≥100 µg/day for 50 years or longer can cause “itai-itai” disease, marked by severe damage to the kidneys and bones, resulting in bone fragility due to osteoporosis and osteomalacia [21,22]. These pathologies have been replicated in ovariectomized cynomolgus monkeys after chronic Cd exposure [23], mimicking the female preponderance of Cd toxicity. Consequently, the kidneys and bones have been employed as the critical targets for Cd toxicity, for which permissible exposure and threshold levels have been determined [24].

As the data in Table 1 indicate, there is no consensus on exposure limits for Cd, even though the same endpoints were used; the dietary Cd exposure limits range between 0.28 and 0.83 μg/kg body weight per day, with Cd exposure threshold levels varying from 1.0 to 5.24 µg/g creatinine for the β_2_M endpoint. Most likely, these inconsistencies are due to different dose–response models, the cut-off values used to define abnormality, and the application of an uncertainty factor or safety margin. A further discussion of these factors is provided in Section 4.4.

Table 2 provides the Cd exposure limits obtained by experimental dosing via oral and inhalational routes. Wu et al. (2012) used data from pigs fed with various doses of Cd in their feed for 100 days, and they reported that the BMDL value for the β_2_M was the highest and it was lowest for the RBP [32]. After the uncertainty factor 100 was applied, a human-tolerable intake level of Cd was established to be 0.2 μg/kg body weight/day.

Based on data from Wistar rats, Faroon et al. (2017) reported the MRL for oral Cd exposure for an intermediate duration (15–365 days) to be 0.5 µg/kg body weight per day for the bone endpoint [33,34,35,36]. Based on data from Fisher rats, the MRL for acute inhalational exposure to Cd for a duration between 1 and 14 days was 0.03 μg/m^3^ for the lung endpoint.

## 3. Imprecisions in Measuring Internal Doses and Adverse Outcomes

The practice of toxicological risk assessment involves measuring two key parameters: exposure and effect indicators. This section focuses on factors which affect the estimation of the internal dose of Cd, which account for the underestimation of the effect size. The uses of blood Cd concentrations and the urinary excretion rates of Cd are highlighted, along with adjusting urine concentrations of Cd and all other indicators of Cd effects to creatinine excretion (E_cr_) and creatinine clearance (C_cr_).

### 3.1. Assimilation of Cd and Its Determinants

From foods, Cd enters the bloodstream through transcytosis [38], receptor-mediated endocytosis [39,40], and specialized transport proteins for essential metals such as iron (Fe), zinc (Zn), and calcium (Ca) [41,42,43,44]. Cd can be expected to be assimilated at a rate higher than Fe, Zn and Ca, consistent with the absorption rate of Cd reported for young Japanese women with low iron intake and low iron status, which was found to be between 24 and 45% [45,46]. Conceivably, the internal dose of Cd in sensitive groups with a higher Cd absorption rate will be more markedly underestimated than in subjects with a normal iron status, in whom the Cd absorption rate is assumed to be 3 to 7%, as in the JECFA’s model [25]. Figure 1 depicts the involvement of enterocytes and metal transport proteins in the assimilation of Cd.

By means of meta-analysis, Peng et al. (2023) found that higher zinc and body iron stores were associated with lower blood Cd concentrations [48]. Higher blood and urine Cd levels in children and adolescent females [49,50] and women of reproductive age [51,52] have been linked to lower body iron stores, as evident from serum ferritin ≤30 µg/L. Like iron, marginal dietary zinc intake and subclinical zinc deficiency are highly prevalent worldwide [53,54,55], which means a significant proportion of the population is at risk of Cd toxicity at the level found in a normal diet.

### 3.2. Use of Blood Cd and Urine Cd in Toxicological Risk Assessment

Through the gut and lungs, Cd in foodstuffs and airborne particle pollution can enter the systemic circulation. Hence, the blood concentration of Cd can reflect recent exposure to the metal. Because most of the Cd in the bloodstream is in the cytosol of red blood cells, which have a 3-month lifespan, blood Cd concentration reflects exposure in the past three months.

As noted in Section 3.1 and reviewed by Cirovic and Cirovic (2024) [56], the amount of Cd that reaches target tissues and organs depends on many factors, which include the absorption rate, nutritional zinc status, and body iron store, not just the amount of Cd in the diet. Consequently, neither blood Cd concentration nor the amount of Cd in the diet can be a precise predictor of an internal dose of Cd and its toxic manifestation. For example, Van Maele-Fabry et al. (2016) reported that breast cancer risk among postmenopausal women was not associated with dietary Cd exposure levels [57]. In comparison, Cd exposure was found to be a strong risk factor for breast cancer in studies in which Cd excretion was used as an exposure indicator [58,59]. Larsson et al. (2015) found that the risk of having breast cancer increased 66% for each 0.5 µg/g creatinine increase in Cd excretion [58]. Lin et al. (2016) reported that breast cancer risk was not associated with dietary Cd exposure, but it was elevated 2.24-fold among women who had Cd excretion rates in the top quartile compared to those with Cd excretion in the lowest quartile [59].

A non-association between blood Cd and diabetes was reported in a recent case–control study from Thailand by Adokwe et al. (2025) [60]. However, in three U.S. population studies, risks of having prediabetes and diabetes were both associated with Cd excretion rates [61,62,63]. In a study by Schwartz et al. (2003), the respective risk of having prediabetes and diabetes rose 48% and 24% at Cd excretion rates of 1–2 μg/g creatinine after smoking and other confounding factors were adjusted [61]. In a study by Wallia et al. (2010), a significant increase in the risk of prediabetes was observed at Cd excretion rates ≥0.7 µg/g creatinine after adjustment for covariates [62]. Jiang et al. (2018) observed that the risk of having prediabetes increased 3.4-fold in obese U.S. men who had a Cd excretion rate in the top quartile compared to those with a normal weight and having Cd excretion rate in the bottom quartile [63]. In summary, the risks of having breast cancer, prediabetes, and diabetes all have been found to be associated with Cd excretion rates lower than 5.24 µg/g creatinine, a threshold level identified when the β_2_M excretion rate ≥300 µg/g creatinine was used as the indicator of an adverse effect of Cd on the kidneys [25].

### 3.3. Urinary Cd Excretion as an Indicator of Body Burden

It is well established that the excretion of Cd can be used as an indicator of cumulative long-term exposure to this metal [64,65]. Specifically, the urine concentration of Cd reflects kidney burden because most of the acquired Cd can be found in the PTCs of the kidneys. These cells release Cd complexed with metallothionein (MT) into the lumen, which then appears in urine following cell damage and cell death from any cause [47].

In human population studies, urine samples are often collected at a single time point (a voided urine sample), and consequently the adjusting of the urine concentrations of Cd and all other excreted biomarkers to creatinine excretion (E_cr_) has been used as a method to correct for differences in urine dilution among people. However, this E_cr_ normalization introduces additional variance to datasets, resulting in the underestimation of the effect size of Cd—further details on this are provided in Section 4. To circumvent such a problem, the normalization of Cd and excreted substances to creatinine clearance (C_cr_) has been used to correct for interindividual differences in urine dilution and the number of functioning nephrons. This C_cr_ normalization has unveiled an unambiguous effect of Cd on eGFR [66] and the excretion of β_2_M, NAG, albumin, and total proteins, which is discussed further in Section 4. The normalization of urinary concentrations of Cd and any excreted substance to E_cr_ and C_cr_ can be undertaken using the equations below.

Excretion of x (E_x_) was normalized to E_cr_ as [x]_u_/[cr]_u_, where x = Cd or any excreted biomarker; [x]_u_ = urine concentration of x (mass/volume); and [cr]_u_ = urine creatinine concentration (mg/dL). E_x_/E_cr_ was expressed as an amount of x excreted per g of creatinine.

Excretion of x (E_x_) was normalized to creatinine clearance (C_cr_) as E_x_/C_cr_ = [Cd]_u_[cr]_p_/[cr]_u_, where x = Cd or any excreted biomarker; [x]_u_ = urine concentration of x (mass/volume); [cr]_p_ = plasma creatinine concentration (mg/dL); and [cr]_u_ = urine creatinine concentration (mg/dL). E_x_/C_cr_ was expressed as an amount of x excreted per volume of the glomerular filtrate [67].

### 3.4. Measurement of Cadmium Effects on Kidneys

As Figure 2 displays, under normal physiological conditions, blood perfuses the kidneys at the rate of 1 L per minute, and all renal blood flow is directed through afferent arterioles into the glomeruli [68]. The plasma entering the glomerulus is filtered into Bowman’s space and 99.9% of the filtered proteins are reabsorbed by tubules, at approximately 40–50 g each day [68].

The protein β_2_M, with a molecular weight of 11,800 Daltons, is expressed on the surface of most nucleated cells and is released into the bloodstream [70,71]. As Figure 2 depicts, β_2_M readily passes through the glomerular membrane to the tubular lumen by virtue of its small mass and is reabsorbed and degraded completely by PTCs [69,70].

As discussed in Section 2, the excretion rate of β_2_M at 300 µg/g creatinine was used as a critical effect of Cd on kidneys [25]. Based on this β_2_M endpoint, dietary exposure to Cd of 58 µg/day was derived as a “tolerable” level. However, a recent study on the homeostasis of β_2_M, as depicted in Figure 2, revealed that the excretion rate of β_2_M is not a reliable indicator of tubular dysfunction [68]. Hence, a suitable endpoint is needed to establish a new health-protective exposure limit for Cd.

It is noteworthy that the risk of having CKD, signified by a decrease in eGFR to one-third of the normal range or is the presence of albuminuria which persists for 3 months or longer [72,73,74], has been linked to a dietary Cd exposure level of 16.7 µg/day [75] and excretion of Cd between 0.27 and 0.37 µg/g creatinine [76,77,78]. Falling eGFR is a clinically relevant outcome of Cd exposure, and it is suggested that the implications of current dietary Cd exposure can be understood in terms of risk for CKD, since both Cd adverse effects and CKD are defined by the same continuous parameter (eGFR loss) [79]. A further discussion of this is provided in Section 4.2.

## 4. Benchmark Dose Modeling of Cd Exposure and Its Nephrotoxicity

This section highlights an application of the BMD methodology to define a point of departure (POD) and a threshold level for Cd exposure in the general population. Its primary aim is to identify the nephrotoxicity endpoint suitable for the estimation of a permissible level of dietary Cd exposure. Its secondary aim is to illustrate the total imprecision which is created by the method used to adjust the urine concentration of Cd and indicators of Cd’s effects on kidneys, such as β_2_M, NAG, albumin, and total protein.

### 4.1. The BMD Softwares for Modeling of Exposure–Effect Datasets

The modeling of exposure–effect data can be performed manually or using dose–response software programs like the PROAST software (https://proastweb.rivm.nl; accessed on 22 March 2025) and the U.S. EPA’s Benchmark Dose Software (BMDS) (https://www.epa.gov/bmds; accessed on 22 March 2025). The application of the PROAST software for continuous and quantal (prevalence) data is exemplified in Section 4.3. The mathematical dose–response models applicable to continuous variables are the inverse exponential, natural logarithmic, exponential, and Hill models [1,3,4,5]. The dose–response models applicable to quantal datasets are the two-stage, logarithmic logistic, Weibull, logarithmic probability, gamma, exponential, and Hill models [1,2].

#### 4.1.1. The BMD Modeling of Continuous Datasets

The BMD modeling of continuous exposure–effect datasets seeks to define the lower bound (BMDL) and upper bound (BMDU) of the 95% confidence interval (CI) of BMD [1]. The lower bound (BMDL) value derived when the benchmark dose response (BMR) is set at 5% could reflect a point of departure (POD) from normality [1,3,4,5]. It is also referred to as the NOAEL equivalent, meaning the level of exposure below which an adverse effect can be discernable. The upper bound (BMDU) is used for computing the BMDU/BMDL ratio, which reflects the uncertainty in the BMD estimates. The greater the difference between the BMDL and BMDU values, the higher the statistical uncertainty in the dataset [1,3,4,5,6].

#### 4.1.2. The BMD Modeling of Quantal (Prevalence) Datasets

The BMD modeling of exposure–outcome prevalence datasets seeks to define the lower bound (BMDL) and upper bound (BMDU) of the 95% confidence interval (CI) of BMD [1,2,6]. The BMDL/BMDU values computed at 5% and 10% prevalence rates of an adverse effect are designated as BMDL_5_/BMDU_5_ and BMDL_10_/BMDU_10_, respectively. The BMDL_5_ can reflect a population threshold level of exposure, defined as an exposure level below which the prevalence of adverse effect can be expected to be ≤5%.

### 4.2. Dose–Response Relationship

In any toxicological risk evaluation, a significant dose–response relationship should first be established. However, as Grandjean and Budtz-Jørgensen (2007) noted, non-differential errors in the measurement of exposure and outcomes, termed total imprecision, can result in a failure to establish a dose–response relationship [19], which would otherwise be established, when such errors are eliminated [19].

As the data in Table 3 indicate, the effect size of Cd on the eGFR was smaller when E_Cd_ was normalized to E_cr_; the doubling of E_Cd_/E_cr_ increased the risk of having low eGFR by 1.47-fold after adjustment for potential confounders [76]. With similar adjustments, the risk of having low eGFR rose 1.96-fold per doubling of E_Cd_/C_cr_.

Table 4 provides additional evidence for non-differential errors created by E_cr_ adjustment [80]. In model A, the risk of having low eGFR was not statistically associated with E_Cd_/E_cr_ (*p* = 0.058), while the risk of having proteinuria rose 3.7-fold as E_Cd_/E_cr_ rose 10-fold (*p* = 0.045).

In model B, the risks of having low eGFR and proteinuria rose 12-fold (*p* < 0.001) and 7-fold (*p* = 0.001), respectively, when there was a 10-fold increase in E_Cd_/C_cr_.

In summary, adjusting urine concentrations of Cd and total protein to E_cr_ generated non-differential errors that could nullify the dose–response relationship. A dose–response relationship could not be established between eGFR and E_Cd_/E_cr_, while the strength of the association between E_Pro_/E_cr_ and E_Cd_/E_cr_ was weak (Table 4). A lack of Cd effects on eGFR has been reported as detailed below.

In a meta-analysis by Jalili et al. (2021), the association of eGFR and E_Cd_/E_cr_ was not significant, while the risk of proteinuria rose by only 35%, when the top category of Cd dose metrics was compared with the bottom Cd exposure category [81]. In an earlier meta-analysis by Byber et al. (2016), there was no evidence for an effect of Cd on eGFR nor progressive eGFR reduction among Cd-exposed individuals [82]. In the latest meta-analysis by Doccioli et al. (2024), a dose–response relationship was observed between eGFR and Cd exposure as there were more data that offset the effect of the non-differential errors caused by E_cr_ normalization [83]. However, there were still insufficient data to connect the risk of albuminuria to Cd exposure [83].

### 4.3. The BMD Modeling Results Using the PROAST Software

Typically, a single or two dose–response models are used in manual BMD computation, which is cumbersome. BMD software programs like the PROAST are increasingly been used as they are freely accessible and offer several advantages; there are many dose–response models to choose and they employ the Akaike information criterion (AIC), which objectively compares the relative goodness of fit of different models [6].

The dose–response curve in which the data best fit offers an insight into the shape and steepness of the slope describing an effect size of Cd. Outputs from the PROAST software applied to continuous and quantal data from the same 409 individuals [80] shown in Table 4 are shown again in Figure 3 and Figure 4.

For the E_Cd_/E_cra_ and E_pro_/E_cr_ datasets (Figure 3), the mathematical dose–response models used were the exponential, Hill, natural logarithmic, and inverse exponential models. Based on the model weights, the exponential model carried the highest weight (0.6840), followed by the Hill model (0.2794), while the natural logarithmic model (0.0386) and inverse exponential model (0.0017) carried much lower weights. According to model averaging, the BMDL value of E_Cd_/E_cr_ associated with a 5% increase in excretion rate of total protein was 0.0536 µg/g creatinine. Notably, the BMDL value of E_Cd_/E_cr_ for an effect on E_pro_ would be unreliable if only Hill model was used in manual BMD modeling.

For the E_Cd_/E_cr_ and proteinuria prevalence datasets (Figure 4A–C), the mathematical dose–response models applied were the two-stage, logarithmic logistic, Weibull, logarithmic probability, gamma, exponential, and Hill models. According to model averaging, the BMDL_5_ value of E_Cd_/E_cr_ for proteinuria was 1.86 µg/g creatinine. The E_Cd_/E_cr_–proteinuria prevalence curve fits the moderately logarithmic probability model (0.3501), followed by the Hill (0.1482) and logarithmic logistic models (0.1452).

The same seven dose–response models were applied to the E_Cd_/E_cr_–CKD prevalence datasets (Figure 4D–F). The BMDL_5_ value of E_Cd_/E_cr_ for CKD was 1.19 µg/g creatinine. The E_Cd_/E_cr_–CKD prevalence curve fits the predominantly exponential model (0.8740), meaning that a small change in E_Cd_/E_cr_ will result in a large increase in CKD prevalence. Thus, the CKD prevalence rate was more sensitive to Cd than proteinuria prevalence.

### 4.4. Reported BMD Values for Different Nephrotoxic Endpoints

The health-based exposure guidance value for Cd exposure, derived from different “POD” figures like MRL, TRV, TWI, TMI, and RfD, assumes that a threshold level of exposure exists [1]. In theory, an exposure level derived from the most sensitive endpoint or the one with the lowest BMDL value would be protective against all other adverse effects [1]. Thus, the lowest BMDL value of E_Cd_/E_cr_ should be used to define the Cd exposure limit. To find the nephrotoxicity endpoint most sensitive to Cd, the results of BMD modeling of continuous and quantal data are summarized in Table 5.

Reported BMDL (BMD) values for Cd excretion rates varied among study populations as well as within the same study populations, depending on the mathematical dose–response models, effect indicators (endpoints), and the cut-off values to define abnormality or deviation from normalcy.

Wang et al. (2016) reported BMDL (BMD) values of E_Cd_/E_cr_ for markers of tubular toxicity, namely RBP, β_2_M, and NAG [85]. Surprisingly few studies have applied the BMD methodology to the data on eGFR, despite this parameter being a diagnostic criterion for CKD. Proteinuria is a hallmark of CKD and predicts continued progressive functional decline of the kidney [88,89,90], but only one paper has performed the BMD modeling of total protein excretion. Interestingly, Cd exposure has been causally related to a rapid fall in eGFR in a prospective cohort study from Switzerland (*n* = 4704) [91]. Respective BMDL_5_ values of E_Cd_/E_cr_ were 1.86 and 1.19 µg/g creatinine when prevalence rates of proteinuria and low eGFR were used as endpoints [80].

It is notable that the BMD values of E_Cd_/E_cr_ for effects on eGFR and tubular injury were marginally different, meaning these two effects are intertwined. A study on Swedish women by Suwazono et al. (2006) reported that the BMDL values of E_Cd_/E_cr_ in Swedish women were 0.5 and 0.7 μg/g creatinine for the tubular injury (E_NAG_/E_cr_) and eGFR endpoints, respectively [84]. Satarug et al. (2022) reported the BMDL values of E_Cd_/E_cr_ for the effects on eGFR, E_NAG_/E_cr_, and E_β2M_/E_cr_ using Thai population data [87]. Hayashi et al. (2024) reported that the BMDL values of E_Cd_/E_cr_ in Japanese women were 3.9 and 3.5 μg/g creatinine for a 10% decrease in the tubular reabsorption of β_2_M and the C_cr_ effect, respectively [86].

Another notable result from Thai population data was that the nephrotoxicity of Cd occurred at a very low body burden. Respective BMDL values of E_Cd_/E_cr_ in men and women were 0.060 and 0.069 µg/g creatinine for the E_NAG_/E_cr_ endpoint [87]. The BMDL value of E_Cd_/E_cr_ was 0.054 µg/g creatinine when a 5% increase in total protein excretion rate was the endpoint [88].

For the quantal (prevalence) data, the prevalence of Cd-related proteinuria increased from 5% to 10% when the population mean value of E_Cd_/E_Cd_ rose from 1.86 to 4.47 µg/g creatinine. In comparison, the prevalence of Cd-related low eGFR increased from 5% to 10% when the population mean value of E_Cd_/E_Cd_ rose from 1.19 to 1.35 µg/g creatinine. Thus, the effect size of Cd on eGFR decline was larger than proteinuria.

### 4.5. Impact of Cadmium Exposure on the Prevalence of CKD 

To demonstrate that the normalization of E_Cd_ and E_alb_ to C_cr_ was superior to a conventional adjustment of urine Cd and urine albumin concentration to E_cr_, the quantal BMD modeling outputs of the PROAST software are summarized in Table 6.

Due to the great difference between BMDU and BMDL, indicated by the BMDU/BMDL ratios ≥ 200, the BMDL_5_ and BMDL_10_ values of E_Cd_/E_cr_ could not be reliably defined for albuminuria. However, the BMDL_5_ and BMDL_10_ values of E_Cd_/E_cr_ could be identified for 5% and 10% CKD prevalence rates. In women, the BMDL_5_ and BMDL_10_ values of E_Cd_/E_cr_ for CKD were 1.93 and 5.31 µg/g creatinine, respectively. The corresponding BMDL_5_ and BMDL_10_ values of E_Cd_/E_cr_ in men were 1.47 and 3.92 µg/g creatinine, respectively. The lower BMDL_5_ and BMDL_10_ values of E_Cd_/E_cr_ in men compared to women were due to higher creatinine excretion rates in men, attributable to a universally higher muscle mass in men than women of a similar age.

For C_cr_-normalized datasets, BMDL_5_ and BMDL_10_ values of Cd exposure levels were determined for both albuminuria and CKD prevalence. For the CKD (low eGFR) prevalence, BMDL_5_ and BMDL_10_ in men and women were not statistically different. Because the basic mechanism of the cytotoxicity of Cd in the PTCs should be the same, the Cd exposure level producing the same toxic outcome can be expected to be identical in men and women. These data strengthen the advantage of C_cr_ normalization of the urine concentration of Cd and albumin. By applying C_cr_ normalization to urine Cd and NAG concentrations, loss of tubular cells per nephron induced by age or Cd was quantified [93].

## 5. Conclusions

The main route of Cd exposure in non-smokers and non-occupationally exposed people is a normal diet. For the kidney target, dosing experiments in pigs identified the oral Cd exposure limit at 0.2 µg/kg body weight per day, while the exposure limits estimated from human data varied from 0.28 and 0.83 µg/kg body weight per day. For the bone target, estimated Cd exposure limits ranged between 0.21 and 0.64 μg/kg body weight/day. These data indicate that kidney and bone toxicity occur at comparable levels of exposure. Therefore, Cd-induced kidney injury may in turn lead to bone injury. The potential kidney–bone connection, even in low-dose exposure conditions, was observed in a Swedish study [94]. In line with this notion is a systematic review and meta-analysis showing a nearly two-fold increase in the risk of having osteoporosis in both low- and high-Cd-exposure groups [95]. In the low-exposure group, the risk of osteoporosis rose 1.95-fold, comparing Cd excretion ≥0.5 versus <0.5 μg/g creatinine. In the high-exposure group, the risk of osteoporosis rose 1.99-fold, comparing Cd excretion ≥5 versus <5 μg/g creatinine [95].

Existing dietary Cd exposure guidelines that were based on β_2_M excretion at a rate above 300 µg/g creatinine as a critical kidney effect are not low enough to be protective of population health, especially in those with low body iron stores and marginal intake of zinc. In low-dose exposure situations, an elevation of β_2_M excretion does not reliably reflect kidney tubular dysfunction. Its use as a basis to estimate an exposure limit for Cd is questionable.

A practice of adjusting the concentration of Cd, β_2_M, NAG, albumin, and total proteins in urine samples to creatinine excretion (E_cr_) creates non-differential errors which bias the dose–response relationship toward the null. It also underestimates or even nullifies the effect size of Cd, especially on eGFR. These impacts of E_cr_ adjustment have resulted in erroneous conclusions made in two meta-analyses that there was no evidence for an effect of Cd on eGFR nor a progressive eGFR decline among Cd-exposed individuals. To eliminate such errors, the urine concentration of Cd and all other indicators of kidney effects should be normalized to creatinine clearance (C_cr_). As another example, the effect of E_Cd_/E_cr_ on the risk of having low eGFR was insignificant, while the risk of having proteinuria rose 3.7-fold per 10-fold increase in E_Cd_/E_cr_. In contrast, the respective risks of having low eGFR and proteinuria rose 12-fold and 7-fold per 10-fold increase in E_Cd_/C_cr_. These results were obtained after adjustment of potential confounders.

Using the excretion of NAG and total protein as indicators of kidney effects, the respective BMDL values for Cd excretion rates were 0.060 and 0.054 µg/g creatinine. These figures are 10-fold below the mean value of Cd excretion rates found in the general populations of many countries, being 0.5–0.6 µg/g creatinine. Alarmingly, these findings imply that the nephrotoxicity of Cd may have occurred already in a significant proportion of people. The incidence of CKD and mortality from CKD will continue to rise at the current dietary Cd exposure levels. As suggested by Ginsberg [79], the effects of environmental Cd exposure on the kidney represent a useful case study of toxicological risk assessment. Arguably, because both Cd’s adverse effects and CKD are defined by the same continuous parameter (eGFR), the implications of exposure limits can be understood in terms of risk for CKD.

The Cd exposure level has now reached toxic levels in a significant proportion of people in many populations, and yet there is no consensus on exposure limits for the metal. Based on BMD modeling data and the fact that there is no theoretical reason to believe that a decrease in eGFR due to nephron destruction by Cd is reversible, new dietary Cd exposure guidelines should be established to preserve kidney functional integrity and to minimize disease progression toward kidney failure.

About 15% of the world’s cultivated land is contaminated with toxic metals, especially Cd, which is prevalent in south and east Asia and parts of the Middle East and Africa [96]. Public health measures should be developed to minimize Cd contamination of food chains and maintain the lowest achievable Cd levels in food crops, especially staples. An effective chelation therapy to remove Cd from the kidneys does not exist. The avoidance of foods containing high Cd levels and smoking cessation are essential preventive measures, as is the maintenance of an optimal body weight and body content of essential metals, notably zinc and iron, to reduce Cd assimilation and kidney burdens to the lowest achievable level.

## Figures and Tables

**Figure 1 toxics-13-00404-f001:**
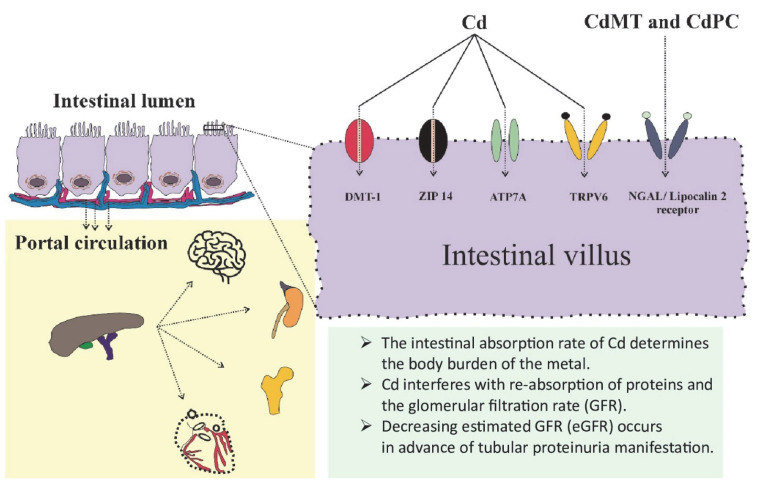
The pathway for cadmium in food to its toxic manifestation in the kidneys. From the gut, Cd can enter the blood stream via specialized transport proteins and pathways for iron, zinc, copper, cobalt, and calcium [41,42,43,44]. Also, the specialized transport proteins for essential metals provide Cd with multiple routes to enter most cells of the body, in which the metal is retained due to a lack of excretory mechanisms. Most of the acquired Cd accumulates within the proximal tubular epithelial cells (PTCs) of the kidneys. The manifestation of toxic Cd accumulation in the kidneys, such as tubulointerstitial inflammation, may incapacitate the glomerular filtration rate. Cd appears in urine following the injury to or death of the PTCs. Via the normalization of Cd excretion (E_Cd_) to creatinine clearance (C_cr_), the amount of Cd exiting the kidneys per nephron can be precisely quantified [47]. Abbreviations: DMT1, divalent metal transporter1; ZIP14, Zrt- and Irt-related protein 10; ATP7A, ATPases (Cu-ATPases); TRPV6, transient receptor potential vanilloid6; NGAL, neutrophil-gelatinase associated lipocalin.

**Figure 2 toxics-13-00404-f002:**
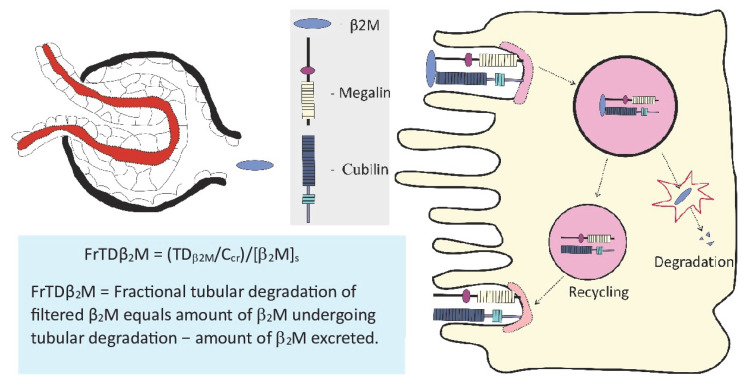
Measuring the tubular effects of cadmium. The reabsorption of β_2_M occurs mostly in the S1 segment of tubules via receptor-mediated endocytosis (RME), involving megalin. Fractional tubular degradation of β_2_M (blue text box), not the excretion rate of β_2_M, has been found to be a reliable indicator of tubular dysfunction, especially in low-dose exposure scenarios [69].

**Figure 3 toxics-13-00404-f003:**
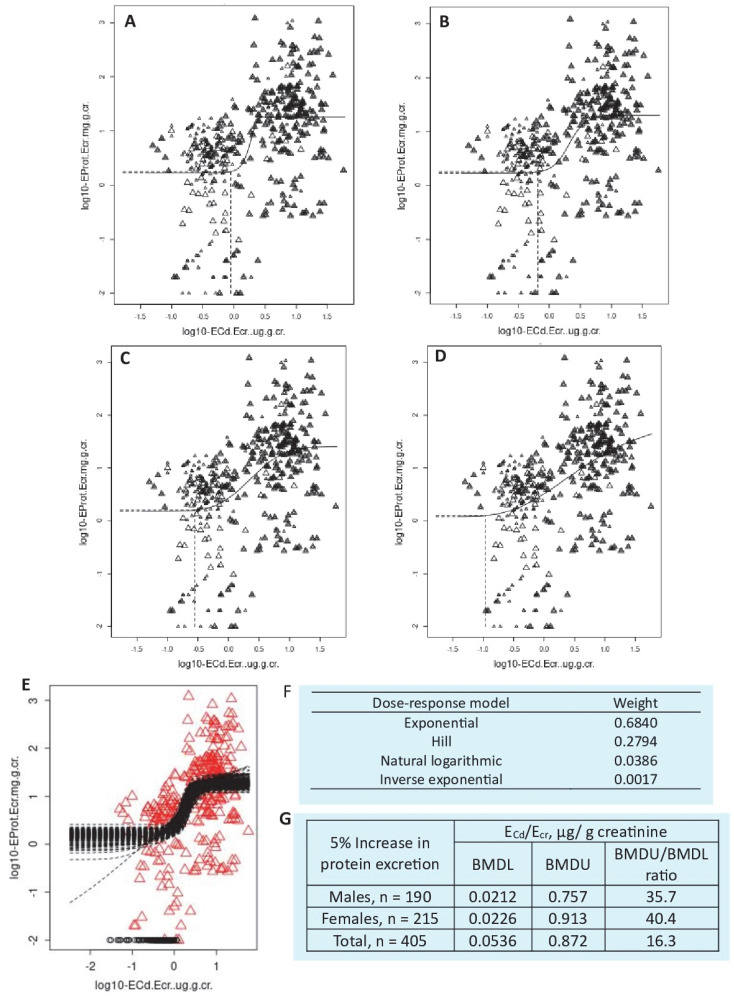
Outputs from the PROAST software applied to E_Cd_/E_cr_ and protein excretion datasets. The weight ranking of each mathematical dose–response model from the highest to the lowest was as follows: exponential (**A**), Hill (**B**), natural logarithmic (**C**), and inverse exponential (**D**). Bootstrap model averaging with 200 repeats (**E**), model weighing (**F**), and BMDL and BMDU values (**G**). Data were from Satarug et al. 2024 [80].

**Figure 4 toxics-13-00404-f004:**
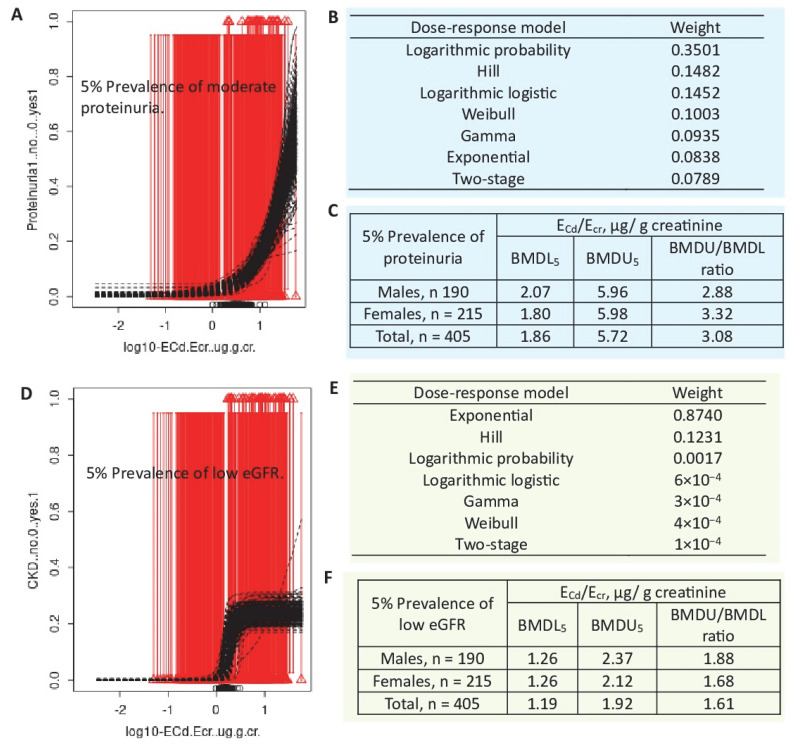
Outputs from the PROAST software applied to E_Cd_/E_cr_–proteinuria and E_Cd_/E_cr_–low eGFR datasets. Bootstrap model averaging with 200 repeats for E_Cd_/E_cr_–-proteinuria (**A**), E_Cd_/E_cr_–-low eGFR (**D**), model weighing (**B**,**E**), and BMDL_5_ and BMDU_5_ values (**C**,**F**). The mathematical dose–response models applied to prevalence datasets are the two-stage, logarithmic logistic, Weibull, logarithmic probability, gamma, exponential, and Hill models. Data were from Satarug et al. 2024 [80].

**Table 1 toxics-13-00404-t001:** Exposure guidelines for Cd in the diet based on kidney and/or bone effects.

Target/Endpoint	Tolerable Intake/Exposure Threshold Level	Reference
Kidneys, β_2_M excretion rate ≥ 300 µg/g creatinine.	A tolerable intake level of 0.83 μg/kg body weight/day (58 µg per day for a 70 kg person). A cumulative lifetime intake of 2 g. Assumed Cd absorption rate of 3–7%. Threshold level of 5.24 μg/g creatinine.	JECFA [25]
Kidneys, β_2_M excretion rate ≥ 300 µg/g creatinine.	A reference dose of 0.36 μg/kg body weight per day (25.2 µg per day for a 70 kg person) Threshold level of 1 μg/g creatinine	EFSA [26,27]
Kidneys, β_2_M and NAG excretion rates	A tolerable intake level of 0.28 μg/kg body weight per day; 16.8 µg/day for a 60 kg person. Threshold levels for the β_2_M and NAG effects were 3.07 and 2.93 μg/g creatinine, respectively. An average dietary Cd exposure in China was 30.6 μg/day.	Qing et al., 2021 [28]
Bones, Bone mineral density	A tolerable Cd intake of 0.64 μg/kg body weight per day. Threshold level of 1.71 μg/g creatinine.	Qing et al., 2021 [29]
Bones, Bone mineral density	A tolerable intake level of 0.35 μg/kg body weight per day. Assumed threshold level of 0.5 μg/g creatinine.	Leconte et al., 2021 [30]
Kidneys and bones, Reverse dosimetry PBPK modeling	Toxicological reference values were 0.21 and 0.36 μg/kg body weight per day, assuming a similar threshold level for effects on kidneys and bones of 0.5 μg/g creatinine.	Schaefer et al., 2023 [31]

NAG, N-acetyl-β-D-glucosaminidase; PBPK, physiologically based pharmacokinetics.

**Table 2 toxics-13-00404-t002:** Exposure guidelines for Cd estimated from dosing experiments.

Target/Animal Species/Dosing Regimes	Exposure Outcome/Exposure Limits	Reference
Kidneys, inbred pigs. Dose: Cd in the feed at 0, 0.5, 2, 8, and 32 mg Cd/kg for 100 days. Cd in tab water was less than 0.001 μg/L.	Tubular dysfunction; urine RBP, NAG, CdMT, and β_2_M. Respective BMDL values of Cd were 0.67, 0.88, 1.00, and 3.08 mg/kg feed for RBP, NAG, CdMT and β_2_M. A tolerable Cd intake level in human was 0.2 μg/kg body weight/day with uncertainty factor 100.	Wu et al., 2012 [32]
Bones, Wistar rats. Dose: CdCl_2_ in drinking water at 0, 1, 5, or 50 mg/L for 6, 9, or 12 months.	Decrease in bone mineral density. Minimal risk level (MRL) for oral Cd in an intermediate exposure duration (15–365 days): 0.5 µg/kg body weight per day	Faroon et al., 2017 [33,34,35,36]
Lungs, Fisher F344 rats Dose: CdO at 0, 0.1, 0.3, 1, 3, or 10 mg CdO/m^3^, for 6.2 h/day, 5 days/week for 2 weeks.	Alveolar histiocytic infiltration and focal inflammation in alveolar septa. MRL for an acute inhalational exposure to Cd for the duration between 1 and 14 days: 0.03 μg/m^3^.	Faroon et al., 2017 [36,37]

BMDL, benchmark dose lower limit; RBP, retinol binding protein; NAG, N-acetyl-β-D-glucosaminidase; CdMT, cadmium complexed with metallothionein; β_2_M, β_2_-microglobulin.

**Table 3 toxics-13-00404-t003:** Effects of the normalization of Cd excretion rate on risk of having low eGFR.

	^a^ Low eGFR			
Model A	POR	95% CI		*p*
		Lower	Upper	
**Log_2_[(E_Cd_/E_cr_) × 10^3^], µg/g creatinine**	1.470	1.276	1.692	<0.001
**Hypertension**	1.632	0.885	3.008	0.117
**Gender**	1.029	0.528	2.002	0.934
**Smoking**	1.232	0.637	2.383	0.536
**BMI, kg/m^2^**				
**12–18**	Referent			
**19–23**	1.058	0.459	2.439	0.894
**≥24**	2.810	1.118	7.064	0.028
**Age, years**				
**16–45**	Referent			
**46–55**	14.23	1.867	108.4	0.010
**56–65**	28.21	3.538	224.9	0.002
**66–87**	141.2	17.87	1116	<0.001
**Model B**	POR	**Lower**	**Upper**	** *p* **
**Log_2_[(E_Cd_/C_cr_) × 10^5^], µg/L filtrate**	1.962	1.589	2.422	<0.001
**Hypertension**	1.735	0.916	3.287	0.091
**Gender**	0.840	0.410	1.719	0.633
**Smoking**	0.944	0.474	1.879	0.869
**BMI, kg/m^2^**				
**12–18**	Referent			
**19–23**	1.109	0.452	2.717	0.822
**≥24**	3.150	1.181	8.400	0.022
**Age, years**				
**16–45**	Referent			
**46–55**	9.951	1.305	75.88	0.027
**56–65**	34.57	4.312	277.2	0.001
**66–87**	198.6	24.59	1605	<0.001

^a^ Low eGFR was defined as eGFR ≤ 60 mL/min/1.73 m^2^. E_Cd_ was normalized to E_cr_ and C_cr_ in models A and B, respectively. Data were from 917 subjects (562 females, 355 males), 16–87 years of age [76].

**Table 4 toxics-13-00404-t004:** Effects of normalization of Cd excretion rate on risks of having low eGFR and proteinuria.

	Low eGFR ^a^		Proteinuria ^b^	
Model A	**POR (95% CI)**	** *p* **	**POR (95% CI)**	** *p* **
**Age, years**	1.121 (1.080, 1.165)	<0.001	1.068 (1.028, 1.110)	0.001
**Log_10_[(E_Cd_/E_cr_) × 10^3^], µg/g creatinine**	2.638 (0.969, 7.182)	0.058	3.685 (1.027, 13.22)	0.045
**Gender**	1.082 (0.490, 2.390)	0.845	1.096 (0.475, 2.528)	0.829
**Smoking**	1.425 (0.596, 3.406)	0.426	1.678 (0.627, 4.486)	0.303
**Hypertension**	2.211 (1.017, 4.805)	0.045	1.113 (0.432, 2.867)	0.824
Model B	**POR (95% CI)**	** *p* **	**POR (95% CI)**	** *p* **
**Age, years**	1.118 (1.073, 1.165)	<0.001	1.061 (1.022, 1.102)	0.002
**Log_10_[(E_Cd_/C_cr_) × 10^5^], mg/L filtrate**	12.24 (3.729, 40.20)	<0.001	7.143 (2.133, 23.92)	0.001
**Gender**	0.802 (0.346, 1.861)	0.608	1.117 (0.482, 2.587)	0.796
**Smoking**	1.335 (0.546, 3.262)	0.527	1.947 (0.725, 5.234)	0.186
**Hypertension**	2.734 (1.204, 6.207)	0.016	1.018 (0.410, 2.530)	0.969

^a^ Low eGFR was defined as eGFR ≤ 60 mL/min/1.73 m^2^. ^b^ Proteinuria was defined as E_pro_/E_cr_ ≥ 100 mg/g creatinine and (E_pro_/C_cr_) × 100 ≥ 100 mg/L filtrate in models A and B, respectively. Data were from 405 subjects (208 females, 197 males) [80].

**Table 5 toxics-13-00404-t005:** BMD modeling of Cd exposure with different nephrotoxicity endpoints.

Endpoints/Population	Results	Reference
NAG and eGFR ***n*** = 790 women, 53–64 years, Sweden	BMDL (BMD) values of E_Cd_/E_cr_ were 0.5 (0.6) and 0.7 (1.1) μg/g creatinine the NAG and eGFR endpoints, respectively.	Suwazono et al., 2006 [84]
RBP, β_2_M and NAG ***n*** = 934 (469 men, 465 women), 10–71 + years, Jiangshan City, Zhejiang, China	BMDL values of E_Cd_/E_cr_ at 5% (10%) BMR in men were 0.89 (1.59), 0.62 (1.30), 0.49 (1.04) μg/g creatinine for the RBP, β_2_M, and NAG endpoints, respectively. Corresponding BMDL values of E_Cd_/E_cr_ in women were 0.76 (1.53), 0.64 (1.34), 0.65 (1.37) μg/g creatinine for the RBP, β_2_M, and NAG endpoints.	Wang et al., 2016 [85]
β_2_M, TRβ_2_M and eGFR (or C_cr_) ***n*** = 112 (Cd-polluted area, ***n*** = 74, non-polluted area, ***n*** = 38) Japan	BMDL values of E_Cd_/E_cr_ in men were 1.8, 1.8, and 3.6 μg/g creatinine for the β_2_M endpoint and decreases in TRβ_2_M by 5% and 10%, respectively. Corresponding BMDL values of E_Cd_/E_cr_ in women were 2.5, 2.6, and 3.9 μg/g creatinine. BMDL values of E_Cd_/E_cr_ for the eGFR (C_cr_) endpoint in men and women were 2.9 and 3.5 μg/g creatinine, respectively	Hayashi et al., 2024 [86]
NAG, β_2_M, and eGFR ***n*** = 734 (Bangkok, ***n*** = 200, Mae Sot, ***n*** = 534), 16–87 years, Thailand	BMDL/BMDU values of E_Cd_/E_cr_ in men were 0.060/0.504 µg/g creatinine for the NAG, while BMDL_10_/BMDU_10_ values were 0.469/0.973 and 3.26/7.46 µg/g creatinine for the β_2_-microglobulinuria and low eGFR ^a^, respectively. Corresponding BMDL/BMDU values of E_Cd_/E_cr_ in women were 0.069/0.537 µg/g creatinine for NAG, while BMDL_10_/BMDU_10_ were 0.733/1.29 and 4.98/9.68 µg/g creatinine for the β_2_-microglobulinuria and low eGFR.	Satarug et al., 2022 [87]
Protein excretion and low eGFR ***n*** = 405 (Bangkok, ***n*** = 100, Mae Sot, ***n*** = 215), 19–87 years, Thailand	BMDL/BMDU values of E_Cd_/E_cr_ for protein loss in men were 0.021/0.757 µg/g creatinine, while BMDL_5_/BMDU_5_ values for proteinuria were 2.07/5.96 µg/g creatinine. Corresponding BMDL/BMDU values of E_Cd_/E_cr_ in women were 0.023/0.913 µg/g creatinine, while BMDL_5_/BMDU_5_ values for proteinuria were 1.80/5.98 µg/g creatinine. In a whole group, BMDL/BMDU values of E_Cd_/E_cr_ for protein loss were 0.054/0.872 µg/g creatinine, while BMDL_5_/BMDU_5_ values were 1.86/5.72 and 1.19/1.92 µg/g creatinine for proteinuria and low eGFR, respectively.	Satarug et al., 2024 [80]

NAG, N-acetyl-β-D-glucosaminidase; eGFR, estimated glomerular filtration rate; RBP, retinal binding protein; β_2_M, β_2_-microglobulin; TRβ_2_M, tubular reabsorption of β_2_M. ^a^ Low eGFR was defined as eGFR ≤60 mL/min/1.73 m^2^, a diagnostic criterion for chronic kidney disease.

**Table 6 toxics-13-00404-t006:** BMDL_5_ and BMDL_10_ of E_Cd_/E_cr_ versus E_Cd_/C_cr_ from albuminuria and CKD prevalence.

Prevalence of Adverse Outcome	E_Cd_/E_cr_, µg/g Creatinine			(E_Cd_/C_cr_) × 100, µg/L Filtrate		
**5% Albuminuria ^a^**	**BMDL_5_**	**BMDU_5_**	**BMDU_5_/BMDL_5_**	**BMDL_5_**	**BMDU_5_**	**BMDU_5_/BMDL_5_**
**Males**	3.06 × 10^−3^	36.7	1.2 × 10^2^	0.163	13	80
**Females**	1.22 × 10^−2^	3.05 × 10^5^	2.5 × 10^7^	0.718	154	60
**10% Albuminuria**	**BMDL_10_**	**BMDU_10_**	**BMDU_10_/BMDL_10_**	**BMDL_10_**	**BMDU_10_**	**BMDU_10_/BMDL_10_**
**Males**	0.55	337	612	1.65	20	12
**Females**	2.52	1.74 × 10^6^	6.7 × 10^5^	3.55	2.12	60
**5% CKD ^b^**	**BMDL_5_**	**BMDU_5_**	**BMDU_5_/BMDL_5_**	**BMDL_5_**	**BMDU_5_**	**BMDU_5_/BMDL_5_**
**Males**	1.47	10.6	7.7	3.22	9.64	2.90
**Females**	1.93	15.6	8.08	3.33	9.20	2.26
**10% CKD**	**BMDL_10_**	**BMDU_10_**	**BMDU_10_/BMDL_10_**	**BMDL_10_**	**BMDU_10_**	**BMDU_10_/BMDL_10_**
**Males**	3.92	15.7	4.00	5.61	13.4	2.39
**Females**	5.31	23.6	4.44	5.88	12.9	2.19

^a^ Albuminuria was defined as urinary albumin-to-creatinine ratios ≥20 mg/g in men and ≥30 mg/g in women for E_cr_-normalized data, while it was defined as (E_Alb_/C_cr_) × 100 ≥ 20 mg/L filtrate in men and ≥30 mg/L filtrate in women for C_cr_-normalized data. ^b^ CKD was defined as eGFR ≤ 60 mL/min/1.73 m^2^. Data were from 603 subjects (203 males, 400 females) [92].

## Data Availability

The original contributions are included in this review. Further inquiries can be directed to the author.

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
