# Peer review of "Challenges in Toxicological Risk Assessment of Environmental Cadmium Exposure"

_toxics, 2025, doi:10.3390/toxics13050404_

Round 1

Reviewer 1 Report

Comments and Suggestions for Authors

This review is well-written.

I hope to incorporate the relationship between renal damage and bone damage after cadmium exposure into this review.

Author Response

Reviewer 1

Comments and Suggestions

This review is well-written.

I hope to incorporate the relationship between renal damage and bone damage after cadmium exposure into this review.

RESPONSES

  • I thank the Reviewer for evaluating and approving my review. Also, I thank the reviewer for raising an important issue on kidney-bone connection, especially in low-dose exposure situations. To the best of my knowledge, no study has yet been conducted to ascertain whether a marker of kidney injury, e.g. GFR, correlates with a marker of bone injury, e.g. bone mineral density in Cd-exposed people.  To stimulate future studies, I have incorporated below paragraph in the Conclusion (lines 413-424) together with additional references. With respect to the English, I have carefully checked for typo errors and undertaken necessary rewordings for clarity.  Changes to the text are in blue.

For kidney target, dosing experiments in pigs identified oral Cd exposure limit at 0.2 µg/ kg body weight per day, while the exposure limits estimated from human data varied from 0.28 and 0.83 µg/ kg body weight per day. For bone target, estimated Cd exposure limits ranged between 0.21 and 0.64 μg/ kg body weight/day. These data indicate that kidney and bone toxicity occur at comparable levels of exposure. Therefore, Cd-induced kidney injury may in turn lead to bone injury. The potential kidney-bone connection even in low-dose exposure conditions was observed in a Swedish study [94]. In line with this notion is a systematic review and meta-analysis showing a nearly two-fold increase in risk of having osteoporosis in both low- and high-Cd exposure groups [95]. In the low-exposure group, risk of osteoporosis rose 1.95-fold, comparing Cd excretion ≥ 0.5 versus < 0.5 μg/g creatinine. In the high-exposure group, risk of osteoporosis rose 1.99-fold, comparing Cd excretion ≥ 5 versus < 5 μg/g creatinine [95].

[94] Wallin, M.; Sallsten, G.; Fabricius-Lagging, E.; Öhrn, C.; Lundh, T.; Barregard, L. Kidney cadmium levels and associations with urinary calcium and bone mineral density: a cross-sectional study in Sweden. Environ. Health 2013, 7, 12, 22.

[95] Kunioka, C.T.; Manso, M.C.; Carvalho, M. Association between Environmental Cadmium Exposure and Osteoporosis Risk in Postmenopausal Women: A Systematic Review and Meta-Analysis. Int. J. Environ. Res. Public Health 2022, 20, 485.

Reviewer 2 Report

Comments and Suggestions for Authors

Below please find my comments the given manuscript.

Comments on manuscript Challenges in Toxicological Risk Assessment of Environmental Cadmium Exposure by Soisungwan Satarug

Introduction

The author points out that the terminology and definition used in risk assessment is not consistent and thus might confuse the reader. It is good that this is brought to attention.

Line 48 change can into cannot

Line 119 ….Japanese women change into young Japanese women with low iron intake and ironstatus..In sensitive groups absorption is higher than in subjects with normal status.

Line 140 3.2 please change into Use of Blood and Urine Cd concentration….

Line 169 break cancer Breast cancer?

Section 3.4 This section describes measuring on tubular reabsorptive function. The authors considers that the fractional tubular degradation of filtered beta2microglobulin is a good indicator of kidney damage. However, it is unclear to the present reviewer how this is measured.

Sections 4.1,4.2, 4.3 The author discuss different models and data software programs for Dose-Response modelling. This section should be scrutinized by a data scientist. Not the competence of the reviewer.

Conclusion

In view of the difficulties to understand the approaches used by the author it is not possible to evaluate if the conclusions are valid.

Author Response

Reviewer 2

Comments and Suggestions

Below please find my comments the given manuscript.

Comments on manuscript Challenges in Toxicological Risk Assessment of Environmental Cadmium Exposure by Soisungwan Satarug

Introduction

The author points out that the terminology and definition used in risk assessment is not consistent and thus might confuse the reader. It is good that this is brought to attention.

RESPONSE: I thank the Reviewer for evaluating my work and for comments to improve a paper.  Necessary revisions have been undertaken to resolve the issues raised.  To illustrate different terms used for Cd exposure limits, a new Table 2 has been inserted (lines 97-99). With respect to the English, I have checked throughout a paper for typo errors, and I have reworded some sentences for clarity.  Changes to the text are in blue.

Comment 1. Line 48 change can into cannot

RESPONSE:

  • For clarity, the word “threshold” has been changed to “a critical exposure level”. The sentence now reads as below.

“The POD-based health guidance values; MRL, TRV, RfD, TWI, and TMI all rely on the premise that a critical exposure level exists, above which an adverse effect cannot be discernable [1].”

Comment 2. Line 119 ….Japanese women change into young Japanese women with low iron intake and ironstatus..In sensitive groups absorption is higher than in subjects with normal status.

RESPONSE: Suggested changes have been undertaken.

Comment 3. Line 140 3.2 please change into Use of Blood and Urine Cd concentration….

RESPONSE:  A change has been undertaken. Thank you for picking up my error.

Comment 4. Line 169 break cancer Breast cancer?

RESPONSE: The error has been corrected to breast cancer.

Section 3.4 .This section describes measuring on tubular re-absorptive function. The authors considers that the fractional tubular degradation of filtered beta2microglobulin is a good indicator of kidney damage. However, it is unclear to the present reviewer how this is measured.

RESPONSE:

  • The equation for determining the fractional tubular degradation of filtered β2M has been provided in the blue text box of Figure 2. It has now been indicated in the legend to Figure 2.

Sections 4.1,4.2, 4.3. The author discussed different models and data software programs for Dose-Response modelling. This section should be scrutinized by a data scientist. Not the competence of the reviewer.

RESPONSE:  

  • To ease an understanding of the BMD modeling, the title of Section 4.1 and Subsections 4.1.1 and 4.1.2 have been changed (see below) and their contents have been simplified.

4.1. The BMD Software for Modeling of Exposure-Effect Datasets

4.1.1. The BMD Modeling of Continuous Datasets

4.1.2. The BMD Modeling of Quantal (Prevalence) Datasets

  • With extensive revisions made, I hope the reviewer now finds that various mathematical dose-response models are integral parts of the PROAST software, which produces the BMDL values for Cd exposure from averaging of all models (lines 304-306). It is the genius of the software developer. As a user, I use the model weights to evaluate different ways of adjusting urine Cd concentrations because the shape and the slope steepness of the dose-response curve can provide unique information on the dataset [ref. 4].

Conclusion

In view of the difficulties to understand the approaches used by the author it is not possible to evaluate if the conclusions are valid.

RESPONSE:

  • The conclusion section has been rewritten and expanded to cover significant issues covered by this review.
  • I hope that you will find the conclusions drawn are well supported by the literatures and data presented in a review.

Reviewer 3 Report

Comments and Suggestions for Authors

The review : "Challenges in Toxicological Risk Assessment of Environmental Cadmium Exposure" by Soisungwan Satarug, summarize evidences of heavy metals pollution , focusing on Cadmium environmental toxicity. Despite the interesting topic, i've spotted a lot of grammatical errors ( i've pointed a few in specific comments) and above all , i suggest the author to be less assertive and more specific (with support of bibliography) with some statements.

Specific comments:  

Lines 8–9:"non-communicable diseases" is a general term; consider specificity, maybe mentioning a few key diseases and drop redundancy with later lines.

Line 11:"Because such a low-dose Cd exposure, results..." is grammatically incorrect. Please revise

Line 94: Inconsistent exposure thresholds across sources not clearly discussed, I suggest the authors to add a comparative synthesis or table commentary.

Line 114: Vague use of “multiple mechanisms” – could name all three directly, it would be better to avoid generalizations and list mechanisms explicitly once.

Line 138:"Experience the nephrotoxicity" is too vague.

Line 183: “Creates a large statistical uncertain” – grammatically incorrect.

Line 212: “Fractional tubular reabsorption...emerged” – lacks citation or support.

Line 232–233: Awkward phrasing: “...for dose-response relationship appraisal.” It’s not clear, please revise.

Line 253–258: Could more clearly explain what BMDL5/BMDL10 represent.

Line 294: Typing error

Line 303: Sentence: “...Cd exposure limits cannot not reliably be derived...”, please revise.

Lines 405–408: I suggest the author to not be overly speculative without caveats, it would be better to add a disclaimer or conditional clause.

Author Response

Reviewer 3.

Comments and Suggestions

The review: "Challenges in Toxicological Risk Assessment of Environmental Cadmium Exposure" by Soisungwan Satarug, summarize evidences of heavy metals pollution, focusing on Cadmium environmental toxicity. Despite the interesting topic, i've spotted a lot of grammatical errors (i've pointed a few in specific comments) and above all, i suggest the author to be less assertive and more specific (with support of bibliography) with some statements.

RESPONSES: I thank the reviewer for evaluating my work and comments to improve a paper.  A paper has undergone extensive revisions to address issues the reviewer raised. With respect to the English, I have checked throughout a paper for typo errors, and I have reworded sentences for clarity. Changes to the text and additional references are in blue. The important changes to the manuscript are listed below.

  • Additional literature reports are provided to support my proposed use of clinically relevant exposure outcomes like falling eGFR, albuminuria and proteinuria in risk assessment for Cd exposure instead of relying on an elevation of β2-microglobulin above 300 µg/g creatinine.
  • A new Table 2 has been inserted to provide Cd exposure limits derived from dosing experiments (lines 97-99).
  • The conclusion section has been rewritten and expanded to cover significant issues covered by this review (lines 411-463).
  • With extensive revisions made, I hope that you will find the conclusions drawn are well supported by the literatures and data presented in a review.

Specific comments:   

Lines 8–9:"non-communicable diseases" is a general term; consider specificity, maybe mentioning a few key diseases and drop redundancy with later lines.

Line 11:"Because such a low-dose Cd exposure, results..." is grammatically incorrect. Please revise

Response: The suggested changes to the abstract have been undertaken as quoted below.

“Dietary exposure to a high-dose cadmium (Cd) ≥ 100 µg/day for at least 50 years or a lifetime intake of Cd ≥ 1 g can cause severe damage to kidneys and bones. Alarmingly, however, exposure to a dose of Cd between 10 and 15 µg/day and excretion of Cd at a rate below 0.5 µg/g creatinine have been associated with increased risk of diseases of high prevalence worldwide such as chronic kidney disease (CKD), fragile bones, diabetes, and cancer. These findings have cast considerable doubt on a “tolerable” Cd exposure of 58 µg/day for a 70 kg person, while questioning a threshold level at Cd excretion rate of 5.24 µg/g creatinine.”

Line 94: Inconsistent exposure thresholds across sources not clearly discussed, I suggest the authors to add a comparative synthesis or table commentary.

Response:  Reasons for inconsistencies have now been provided (lines 91-93), quoted below.

“Most likely, these inconsistencies are due to different dose-response models, the cut-off values to define abnormality and an application of uncertainty factor or safety margin. A further discussion is in Section 4.4.”

Line 114: Vague use of “multiple mechanisms” – could name all three directly, it would be better to avoid generalizations and list mechanisms explicitly once.

Response: The term “multiple mechanisms” has been deleted.

Line 138:"Experience the nephrotoxicity" is too vague.

Response:  The referred sentence has been changed to read as below.

“Like iron, marginal dietary zinc intake and subclinical zinc deficiency are highly prevalent worldwide [53-55], which means a significant proportion of population is at risk of Cd toxicity at the level found in a normal diet.”

Line 183: “Creates a large statistical uncertain” – grammatically incorrect.

Response: It has been changed to “introduces additional variance to datasets”

Line 212: “Fractional tubular reabsorption...emerged” – lacks citation or support.

Response: The referred sentence has been replaced with a new sentence with a reference, quote bleow

“Fractional tubular degradation of β2M (blue text box), not the excretion rate of β2M, has been found to be a reliable indicator of tubular dysfunction, especially in low-dose exposure scenarios [68].

[68] Phelps, K.R.; Yimthiang, S.; Pouyfung, P.; Khamphaya, T.; Vesey, D.A.; Satarug, S. Homeostasis of β2-Microglobulin in Diabetics and Non-Diabetics with Modest Cadmium Intoxication. Scierxiv 2025, 2025, 60. https://doi.org/10.20517/scierxiv.2025.60.v1

Line 232–233: Awkward phrasing: “...for dose-response relationship appraisal.” It’s not clear, please revise.

Response: The referred sentence has been replaced by “The BMD Software for Modeling of Exposure-Effect Datasets”

Line 253–258: Could more clearly explain what BMDL5/BMDL10 represent.

Response: The definitions for BMDL5/BMDL10 are provided in Section 4.1.2. The BMD Modeling of Quantal (Prevalence) Datasets (lines 258-264).

In addition, Table 6 provides BMDL5/BMD10 values of Cd excretion based on the prevalence rates of albuminuria and CKD. The interpretations of these figures are provided also.

Line 294: Typing error

Response: A correction has been undertaken.

Line 303: Sentence: “...Cd exposure limits cannot not reliably be derived...”, please revise.

Response: A correction has been undertaken.

Lines 405–408: I suggest the author to not be overly speculative without caveats, it would be better to add a disclaimer or conditional clause.

Response: Thank you for the caution advice.

Round 2

Reviewer 1 Report

Comments and Suggestions for Authors

can accept